# Ritonavir Has Reproductive Toxicity Depending on Disrupting PI3K/PDK1/AKT Signaling Pathway

**DOI:** 10.3390/toxics12010073

**Published:** 2024-01-15

**Authors:** Eun-Ju Jung, Jae-Hwan Jo, Claudine Uwamahoro, Seung-Ik Jang, Woo-Jin Lee, Ju-Mi Hwang, Jeong-Won Bae, Woo-Sung Kwon

**Affiliations:** 1Department of Animal Science and Biotechnology, Kyungpook National University, Sangju 37224, Gyeongsangbuk-do, Republic of Korea; red0787@naver.com (E.-J.J.); claudineuwa20@gmail.com (C.U.); todwnl5787@naver.com (S.-I.J.); wj9059lee@naver.com (W.-J.L.); ghkdwnal100@gmail.com (J.-M.H.); jwbae1822@gmail.com (J.-W.B.); 2Department of Animal Biotechnology, Kyungpook National University, Sangju 37224, Gyeongsangbuk-do, Republic of Korea; ocjallk@naver.com; 3Research Institute for Innovative Animal Science, Kyungpook National University, Sangju 37224, Gyeongsangbuk-do, Republic of Korea

**Keywords:** ritonavir, spermatozoa, capacitation, PI3K/PDK1/AKT pathway

## Abstract

Ritonavir (RTV) is an antiviral and a component of COVID-19 treatments. Moreover, RTV demonstrates anti-cancer effects by suppressing AKT. However, RTV has cytotoxicity and suppresses sperm functions by altering AKT activity. Although abnormal AKT activity is known for causing detrimental effects on sperm functions, how RTV alters AKT signaling in spermatozoa remains unknown. Therefore, this study aimed to investigate reproductive toxicity of RTV in spermatozoa through phosphoinositide 3-kinase/phosphoinositide-dependent protein kinase-1/protein kinase B (PI3K/PDK1/AKT) signaling. Duroc spermatozoa were treated with various concentrations of RTV, and capacitation was induced. Sperm functions (sperm motility, motion kinematics, capacitation status, and cell viability) and expression levels of tyrosine-phosphorylated proteins and PI3K/PDK1/AKT pathway-related proteins were evaluated. In the results, RTV significantly suppressed sperm motility, motion kinematics, capacitation, acrosome reactions, and cell viability. Additionally, RTV significantly increased levels of phospho-tyrosine proteins and PI3K/PDK1/AKT pathway-related proteins except for AKT and PI3K. The expression level of AKT was not significantly altered and that of PI3K was significantly decreased. These results suggest RTV may suppress sperm functions by induced alterations of PI3K/PDK1/AKT pathway through abnormally increased tyrosine phosphorylation. Therefore, we suggest people who use or prescribe RTV need to consider its male reproductive toxicity.

## 1. Introduction

Ritonavir (RTV) is one of the antiviral medicines belonging to the protease inhibitor class, which is used for controlling HIV (human immunodeficiency virus) [1]. The eruption of COVID-19 necessitated the development of new medicine and repurposing of drugs [2,3,4]. Currently, RTV is used as one of the components of commercially available treatment for COVID-19 [5] and is also known for its anti-cancer effects [6,7]. RTV induced dephosphorylation of AKT leading to apoptosis in breast cancer cells [6], and significantly suppressed cell proliferation by inhibiting AKT phosphorylation in pancreatic cancer cells [7]. Although RTV has been broadly used, it was known for its several side effects [8,9]. RTV reduces cell viability and increases cytotoxicity, mitochondria DNA damage, and apoptosis in human endothelial cells [8]. In human epithelium kidney cells, RTV induces cytotoxicity, endoplasmic reticulum stress, and mitochondria stress [9]. Especially, RTV suppressed mouse sperm function including cell viability, capacitation, and acrosome reaction at concentrations lower than those toxic to other cells [10].

The ejaculated sperm go through a capacitation in the female reproductive tract to acquire the ability for successful sperm-egg fusion [11,12]. During capacitation, sperm motility and diverse molecular functions, including protein kinase A and tyrosine-phosphorylated substrates, are altered. Following this procedure, only spermatozoa that are capacitated can undergo the acrosome reaction, and only acrosome-reacted spermatozoa can penetrate the oocyte [13]. The phosphoinositide 3-kinase/phosphoinositide-dependent protein kinase-1/protein kinase B (PI3K/PDK1/AKT) pathway has been reported to be changed and regulate sperm functions during the capacitation process. PI3K should be fully activated toward the end of sperm capacitation [14]. Therefore, inhibition of PI3K during capacitation decreases the level of phosphorylated AKT and suppresses sperm motility, vitality, and acrosome reaction [15,16].

While a clue is provided regarding the effect of RTV on sperm function through AKT phosphorylation, the mechanism by which RTV alters the PI3K/PDK1/AKT pathway-related proteins is not known. Thus, this study aimed to explore the effects of RTV on sperm functions and to identify activation change of PI3K/PDK1/AKT pathway-related proteins by RTV treatment. This study is expected to contribute to elucidating the molecular mechanism of how RTV affects male fertility.

## 2. Materials and Methods

All process were executed in strict compliance with the instructions for the ethical treatment of animals as permitted by the Institutional Animal Care and Use Committee of Kyungpook National University.

### 2.1. Media and Chemicals

The media were prepared following the previous study [17]. Modified tissue culture medium 199 [2.92 mM calcium lactate, 3.05 mM D-glucose, 0.91 mM sodium pyruvate, 10%, 2.2 g/L sodium bicarbonate, and fetal bovine serum (Sigma-Aldrich, St. Louis, MO, USA)] was used as the basic medium (BM). To induce the capacitation, capacitation media (CM), the addition of 10 μg/mL heparin to BM, was used [17].

### 2.2. Sample Preparation

Male durocs were housed at a temperature-controlled and ventilated facility in Gyeongsan Swine Gene (Gyeongsan, Republic of Korea). The semen samples of Duroc aged 24 to 36 months were gathered by gloved-hand method and diluted to a density of 3 × 10^5^ sperm cells/mL with a Beltsville thawing solution. The diluted semen samples were kept at 17 °C and analyzed within 2 h of collection. Five semen samples were randomly mixed to avoid individual factors in fertility between sperm samples for each replicate following the previous study [18]. The procedure of semen washing was followed as described in previous study [19]. In brief, dead sperm and seminal plasma were removed by washing with a discontinuous (35% [*v*/*v*] and 70% [*v*/*v*]) Percoll gradient (Sigma-Aldrich, St. Louis, MO, USA) at 500× *g* for 20 min. Then, samples were washed again with BM at 500× *g* for 5 min to eliminate remaining Percoll. Finally, samples were incubated with CM containing 0, 0.1, 1, 10, and 100 μM of RTV for 60 min under the condition of 37 °C and 5% CO_2_ to induce capacitation [18]. The RTV concentrations were established by referring to previous studies [10,20,21].

### 2.3. Sperm Motility and Motion Kinematics

The sperm motility and motion kinematics were analyzed according to previous studies [18,19]. The following parameters were measured: sperm motility [total sperm motility (MOT, %), rapid sperm motility (RPD, %), medium sperm motility (MED, %), slow sperm motility (SLW, %), and progressive sperm motility (PRG, %)] and motion kinematic parameters [curvilinear velocity (VCL, μm/s), straight-line velocity (VSL, μm/s), average path velocity (VAP, μm/s), linearity (LIN, %), straightness (STR, %), beat cross frequency (BCF, Hz), mean angular displacement (MAD, degree), wobble (WOB, %), dance (DNC, μm^2^/s), dance mean (DNM, μm), and amplitude of lateral head displacement (ALH, μm)].

### 2.4. Sperm Capacitation Status

The capacitation status of spermatozoa was measured by the dual staining method [combined chlortetracycline (CTC) fluorescence/Hoechst 33258 (H33258) evaluation] as described previously [22]. After the staining, at least 400 spermatozoa were classified into four patterns according to their capacitation status: dead (D pattern, blue fluorescence on the sperm head), live non-capacitated (F pattern, bright green fluorescence spread evenly on the sperm head), live capacitated (B pattern, bright green fluorescence in acrosomal region), or acrosome-reacted (AR pattern, no fluorescence on the sperm head or bright green fluorescence only in the post acrosomal region). The representative images of each pattern were presented in Appendix A.

### 2.5. Cell Viability

The cell viability was measured following the procedure of a previous study [19]. A cell viability assay kit (Abcam, Cambridge, UK) was utilized according to the instructions of manufacturer.

### 2.6. Western Blotting Analysis

Western blot analysis was conducted to quantity the expression levels of PI3K/PDK1/AKT signaling pathway-related proteins [PI3K, phospho-PI3K (Tyr^607^), AKT, phospho-AKT (Thr^308^ and Ser^473^), phosphoinositide-dependent kinase-1 (PDK1), phospho-PDK1 (Ser^241^)phosphatase and tensin homolog (PTEN), and phospho-PTEN] and tyrosine-phosphorylated proteins, following the procedure described in previous studies [23,24]. In brief, after the samples were washed twice with DPBS, pellets were resuspended in modified Laemmli sample buffer (10% SDS, 10% glycerol, 5% 2-mercaptoethanol, 315 mM Tris, and 5% bromophenol blue) and incubated for 10 min at RT. The samples were centrifuged at 10,000× *g* for 7 min before the supernatants were boiled at 95 °C for 3 min. The protein samples were divided by 12% SDS-PAGE (MiniPROTEIN Tetra Cell, Bio-Rad, Hercules, CA, USA) before they were transferred to PVDF membranes (Bio-Rad). The membranes were incubated in 3% ECL blocking agent (GE Healthcare, Chicago, IL, USA) for 2 h at RT. The membranes were further incubated with diluted primary antibodies in 3% ECL blocking agent [anti-PI3K antibody (Proteintech Group, Inc., Rosemont, IL, USA), 1:20,000; anti-phospho-PI3K (Tyr^607^) antibody (Affinity Biosciences, Cincinnati, OH, USA), 1:2000; anti-AKT antibody (Cell Signaling Technology, Danvers, MA, USA), 1:1000; anti-phospho-AKT (Thr^308^) antibody (Cell Signaling Technology, Danvers, MA, USA), 1:1000; anti-phospho-AKT (Ser^473^) antibody (Genetex, Inc., Irvine, CA, USA), 1:1000; 1: 1000; anti-PDK1 antibody (Biorbyt, Ltd., Cambridge, UK), 1:1000; anti-phospho-PDK1 (Ser^241^) antibody (LSBio, Inc., Seattle, WA, USA), 1:1000; anti-PTEN antibody (MyBioSource, Inc., Sandiego, CA, USA), 1:1000; anti-phospho-PTEN antibody (Bioss, Inc., Woburn, MA, USA), and anti-phosphotyrosine antibody [PY20] (HRP) (Abcam, Cambridge, UK), 1:5000; anti-alpha tubulin antibody (Abcam), 1:5000]. The membranes were washed and incubated with secondary antibodies. Goat anti-rabbit IgG H&L (HRP) (Cell Signaling Technology, Danvers, MA, USA) and goat anti-mouse IgG H&L (HRP) (Abcam, Cambridge, UK) were diluted with 3% ECL blocking agent to 1:2000 and used as secondary antibodies. Proteins on membranes were probed by the enhanced chemiluminescence method using the iBright™ CL1500 Imaging System (ThermoFisher Scientific, Waltham, MA, USA). Image Studio Lite system (Version 5.2, LI-COR Corporate, Lincoln, NE, USA) was utilized for measuring protein expression levels. The density ratio of each treatment band was standardized to match the α-tubulin ratio.

### 2.7. Statistical Analysis

One-way analysis of variance (ANOVA) was used to test the data for significant differences among treatments. To compare the four RTV treatment groups with the control group, Tukey’s post hoc test was utilized. All statistical analyses were performed in SPSS software (version 26.0, IBM, Armonk, NY, USA). Each experiment was replicated at least three times. Significant differences were defined as those with a *p*-value of less than 0.05 (*p* < 0.05). The data were presented mean ± SEM. 

## 3. Results

### 3.1. The Impact of Ritonavir on Sperm Motility and Motion Kinematics

MOT, RPD, and PRG were significantly decreased at the highest concentration of RTV (100 μM) in a dose-dependent manner (*p* < 0.05, Table 1). In addition, VCL, VSL, VAP, LIN, BCF, MAD, DNC, DNM, and ALH were also significantly decreased in response to the highest concentration of RTV (100 μM) (*p* < 0.05, Table 1). However, MED, SLW, STR, and WOB did not show significant differences (Table 1).

### 3.2. The Impact of Ritonavir on Capacitation Status and Cell Viability

The ratio of the AR pattern was significantly reduced at high concentrations of RTV (10 and 100 μM), and the ratio of the B pattern was significantly decreased from 1 μM of RTV dose-dependently (*p* < 0.05, Figure 1A,B). In contrast, the ratio of the F pattern significantly increased dose-dependently starting at 1 μM of RTV (*p* < 0.05, Figure 1C). The cell viability was significantly decreased at high concentrations of RTV (10 and 100 μM) in a dose-dependent manner (*p* < 0.05, Figure 1D).

### 3.3. Expression Levels of Tyrosine-Phosphorylated Proteins and PI3K/PDK1/AKT Pathway-Related Proteins

Tyrosine-phosphorylated protein was detected at approximately 65 kDa (Figure 2B). RTV treatment significantly increased tyrosine-phosphorylated protein level at high concentrations (10 and 100 μM) (*p* < 0.05, Figure 2A). AKT and phospho-AKT (Thr^308^) were detected at approximately 60 kDa, and phospho-AKT (Ser^473^) was detected at approximately 60 and 58 kDa (Figure 3E). There was no significant change in the expression level of AKT. However, the expression level of phospho-AKT (Thr^308^) was significantly increased at the highest concentration of RTV (100 μM). The expression level of phospho-AKT (Ser^473^) approximately 60 kDa band was significantly increased at the highest concentration of RTV (100 μM), and approximately 58 kDa band was significantly increased at high concentrations (10 and 100 μM) (*p* < 0.05, Figure 3A). PI3K and phospho-PI3K were detected at approximately 85 kDa (Figure 3E). RTV treatment significantly decreased the expression level of PI3K at the highest concentration (100 μM) but significantly increased that of phospho-PI3K starting at 1 μM (*p* < 0.05, Figure 3B). PTEN and phospho-PTEN were detected at approximately 55 kDa (Figure 3E). The expression level of PTEN was significantly increased at high concentrations (10 and 100 μM) and phospho-PTEN was significantly increased from 1 μM (*p* < 0.05, Figure 3C). PDK1 and phospho-PDK1 were detected at approximately 65 kDa (Figure 3E). The expression level of PDK1 was significantly increased at the highest concentration (100 μM) and phospho-PDK1 was significantly increased at high concentrations (10 and 100 μM) dose-dependently by RTV treatment (*p* < 0.05, Figure 3D). 

## 4. Discussion

RTV, which is used to control HIV and is now also used as a component of COVID-19 treatment, boosts up another reagent nirmatrelvir [25]. Additionally, RTV has anti-cancer effects [7,26,27]. However, RTV was cytotoxic and even toxic to the male reproductive system at concentrations lower than those previously reported to be toxic in other cells [20,28]. In spermatozoa, RTV decreases motility, motion kinematics, capacitation status, acrosome reaction, and cell viability. This effect is due to abnormal tyrosine phosphorylation and AKT level [10]. However, how RTV regulates the PI3K/PDK1/AKT pathway is not yet known. Therefore, this study sought to evaluate how RTV treatment influenced sperm functions by assessing signaling transduction through observing expression levels of PI3K/PDK1/AKT pathway-related proteins and their phosphorylated forms by Western blotting. 

In the present study, boar sperm functions were suppressed by RTV as in a previous experiment with mouse spermatozoa [10]. Sperm motility and motion kinematics were significantly suppressed (Table 1). Additionally, live-capacitated spermatozoa were significantly decreased. Whereas a previous study reported an increased acrosome reaction in mouse sperm, in this study, it was decreased in boar sperm (Figure 1A). This might be due to the species differences. Cell viability was significantly decreased at high concentrations (10 and 100 μM) of RTV, similar to a previous study with mouse spermatozoa (Figure 1D) [10]. This result indicates RTV may have cytotoxicity in sperm cells starting at 10 μM. The results of our previous study [10] and the present study indicate RTV causes adverse effects on sperm functions.

In addition to suppression of sperm function, the expression levels of the PI3K/PDK1/AKT pathway-related proteins were significantly altered by RTV. Generally, PI3K phosphorylates phosphatidylinositol (4,5)-bisphosphate (PIP_2_) to phosphatidylinositol (3,4,5)-trisphosphate (PIP_3_), whereas PTEN dephosphorylates the PIP_3_ to PIP_2_ in the PI3K/PDK1/AKT pathway [29]. AKT is attached to PIP_3_ and is activated by PDK1, which phosphorylates the Thr^308^ residue of AKT [30]. Additionally, the Ser^473^ residue needs to be phosphorylated for inducing the full activation of AKT [31]. In this study, the level of PI3K was significantly decreased whereas that of p-PI3K was significantly increased by RTV treatment. The levels of PTEN and p-PTEN were both significantly increased (Figure 3B,C). PI3K and PTEN are involved in sperm viability and motility [32,33,34]. In boar spermatozoa, inhibition of PI3K by LY294002 or wortmannin, which competed with ATP for binding the PI3K active site, significantly decreased cell viability but not capacitation status or acrosome reaction under capacitating conditions [35]. In mouse and human spermatozoa, total and progressive motility were significantly decreased by wortmannin [36]. Additionally, treatment with Ochratoxin A activated PTEN and deactivated AKT, which reduced sperm motility [37]. The increase in reactive oxygen species caused by the air pollutant hydrogen sulfide similarly activated PTEN, resulting in the inhibition of AKT, which in turn led to a decrease in sperm motility [32]. Similarly, in the present study, sperm motility and motion kinematics were significantly suppressed at the highest concentration of RTV (100 μM) (Table 1), and cell viability was significantly decreased (Figure 1D). We, therefore, conclude RTV increases the phosphorylation of PI3K and PTEN leading to the decline in sperm functions.

Moreover, there was no significant alteration of the AKT level whereas p-AKT (Thr^308^ and Ser^473^) expression levels were significantly increased by RTV treatment. PDK1 and p-PDK1 expression levels were significantly increased as well. In a previous study, AKT inhibition with SH-5 in stallion spermatozoa decreased total and progressive motility, motion kinematics, and membrane integrity [38]. In another previous research, AKT inhibition with AKTi-2 in mouse spermatozoa during capacitation significantly reduced total and progressive motility, tyrosine phosphorylation, and acrosome reaction, whereas there was no significant change in cell viability [39]. In regards to the previous study on PDK1 in boar spermatozoa, inhibition of PDK1 phosphorylation at the site of Ser^241^ increased tyrosine phosphorylation [40]. Similarly, soon after inducing full hyperactivation of boar spermatozoa, PDK1 abruptly became inactivated by dephosphorylation in the Ser^241^ region, and protein tyrosine phosphorylation was increased [41]. Similarly, MOT, PRG, VCL, VSL, and VAP were significantly suppressed at the highest concentration of RTV (100 μM) in this study. Additionally, the acrosome reaction was significantly diminished and tyrosine-phosphorylated substrate level was significantly increased. These results imply PI3K and PTEN signaling alteration by RTV increased the expression of PDK1 and p-PDK1 followed by increased AKT phosphorylation. Finally, this series of signaling changes suppress physiological sperm functions. However, further research is needed to establish the exact causal relationship between altered expression level of proteins and suppressed sperm function.

In a previous study, RTV induced abnormal phosphorylation of AKT and tyrosine substrates in mouse spermatozoa [10]. However, the mechanism of these molecular changes was unclear. Therefore, the PI3K/PDK1/AKT signaling pathway-related proteins expression levels were additionally measured in this study. PI3K has tyrosine residues including Tyr^580^, Tyr^607^, and Tyr^688^. Phosphorylation of Tyr^688^ in the CSH2 domain increases activity of the catalytic domain of PI3K [42], resulting in transition of PIP_2_ to PIP_3_ followed by adhesion of AKT and PDK1 to the plasma membrane [29]. Activated PDK1 phosphorylates Thr^308^ residue of AKT in sequence [30]. The association between PDK1 and tyrosine phosphorylation has been reported in several previous studies [43,44,45]. PDK1 has tyrosine residues Tyr^9^, Tyr^373^, and Tyr^376^. In HEK 293 cells, these residues are phosphorylated for further activation of PDK1 after the phosphorylation of Ser^241^ [45]. The tyrosine phosphorylation of PDK1 most likely increases the stabilization of the bond between PDK1 and the membrane [44]. Additionally, phosphorylation of Tyr^373^ and Tyr^376^ might increase catalytic activity of PDK1 [43]. Moreover, Harayama et al. (2008) demonstrated phosphorylation of PDK1 was increased with tyrosine phosphorylation. Hyperactivation inactivates PDK1 by dephosphorylating Ser^241^, and the tyrosine-phosphorylated proteins are increased simultaneously in boar spermatozoa [41]. Therefore, we propose RTV alters the PI3K/PDK1/AKT signaling by abnormal tyrosine phosphorylation (Figure 4). In Figure 4, the PI3K/PDK1/AKT pathway affected by RTV was depicted. RTV induced abnormal protein tyrosine phosphorylation. In addition, phosphorylation of PI3K/PDK1/AKT pathway-related proteins were increased. Since PI3K and PDK1 have tyrosine residues, they may be affected by increased tyrosine residues [41,42,44]. Moreover, PDK1 activation could affect the protein tyrosine phosphorylation [41]. Therefore, the alteration of PI3K/PDK1/AKT signaling by RTV may suppress the sperm functions.

## 5. Conclusions

The present study is the first to investigate the effect of RTV on PI3K/PDK1/AKT pathway in spermatozoa. In addition, this study can support previous research on the detrimental effects of RTV on sperm and provide data on its molecular mechanism. Our results showed treatment of RTV during capacitation suppressed sperm functions. In addition, RTV-induced molecular changes were observed. It was expected that increased aberrant tyrosine phosphorylation may increase the phosphorylation of PI3K and PDK1, which may enhance AKT phosphorylation (Figure 4). Taken together, RTV is predicted to have reproductive toxicity by disrupting PI3K/PDK1/AKT signaling pathway. Therefore, we suggest when using or administering RTV, particular attention be given to male reproductive toxicity. The findings of this study are anticipated to support the further investigation of PI3K/PDK1/AKT signaling.

## Figures and Tables

**Figure 1 toxics-12-00073-f001:**
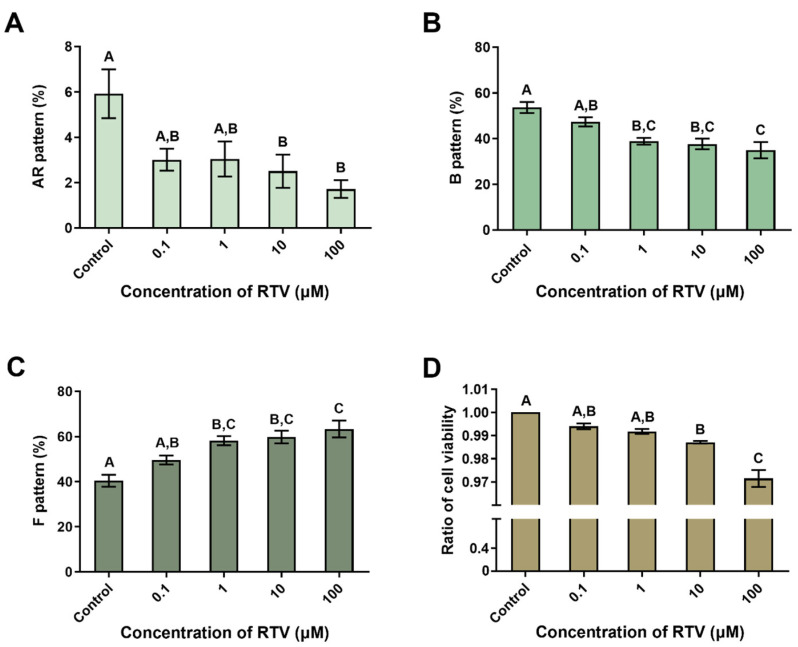
Impacts of ritonavir (RTV) on capacitation status and cell viability. (**A**) Live acrosome-reacted pattern (AR pattern). (**B**) Live capacitated pattern (B pattern). (**C**) Live non-capacitated pattern (F pattern). (**D**) Cell viability levels after treatment with various RTV concentrations (0, 0.1, 1, 10, and 100 µM). Values with different superscripts (^A^, ^B^, and ^C^) indicate significant differences between the control and treatment groups (*p* < 0.05). Data were analyzed by one-way ANOVA and expressed as mean ± SEM; n = 8 (**A**–**C**) and n = 3 (**D**).

**Figure 2 toxics-12-00073-f002:**
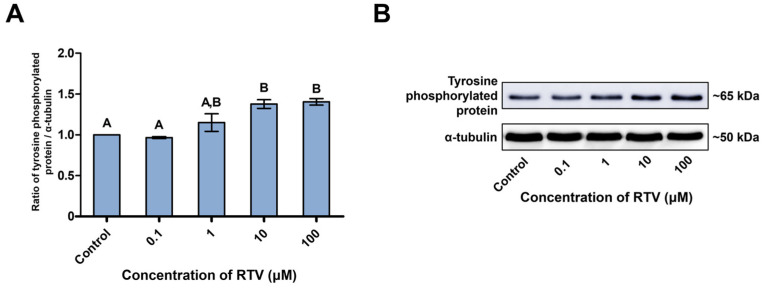
Impact of ritonavir (RTV) on protein tyrosine phosphorylation. (**A**) The level of tyrosine-phosphorylated protein measured at approximately 65 kDa. (**B**) Probing of the tyrosine-phosphorylated protein. Lane 1: Control; lane 2: 0.1 µM RTV; lane 3: 1 µM RTV; lane 4: 10 µM RTV; lane 5: 100 µM RTV. Data were analyzed by one-way ANOVA and expressed as mean ± SEM, n = 3. Values with different superscripts (^A^ and ^B^) indicate significant differences between the treatment groups and control (*p* < 0.05).

**Figure 3 toxics-12-00073-f003:**
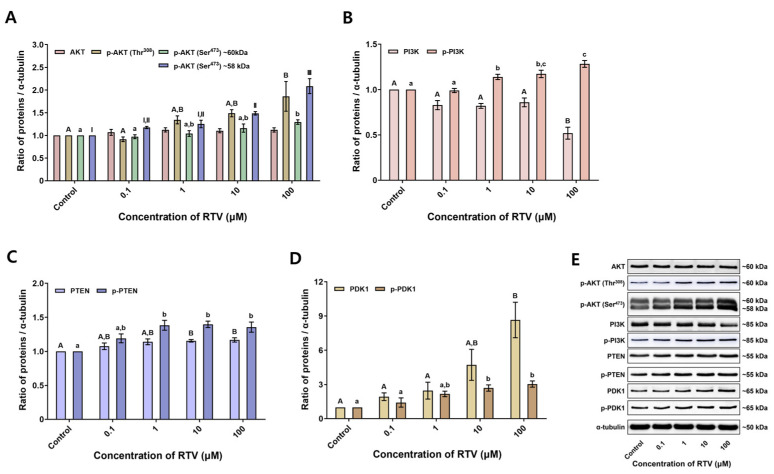
Impacts of ritonavir (RTV) on levels of PI3K/PDK1/AKT pathway proteins. (**A**) AKT and phosphorylated AKT [p-AKT (Thr^308^)] levels were measured at approximately 60 kDa, and phosphorylated AKT [p-AKT (Ser^473^)] level was measured at approximately 60 and 58 kDa. (**B**) Phosphoinositide 3-kinase (PI3K) and phosphorylated PI3K levels were measured at approximately 85 kDa. (**C**) Phosphatase and tensin homolog (PTEN) and phosphorylated PTEN levels were measured at approximately 55 kDa. (**D**) Phosphoinositide-dependent kinase-1 (PDK1) and phosphorylated PDK1 levels were measured at approximately 65 kDa. (**E**) Probing of the PI3K/PDK1/AKT pathway proteins. Lane 1: Control; lane 2: 0.1 µM RTV; lane 3: 1 µM RTV; lane 4: 10 µM RTV; lane 5: 100 µM RTV. Data were analyzed by one-way ANOVA and expressed as mean ± SEM; n = 3. Values with different superscripts (^A^, ^B^, ^a^, ^b^, ^c^, ^I^, ^II^, and ^III^) indicate significant differences between the treatment groups and control (*p* < 0.05).

**Figure 4 toxics-12-00073-f004:**
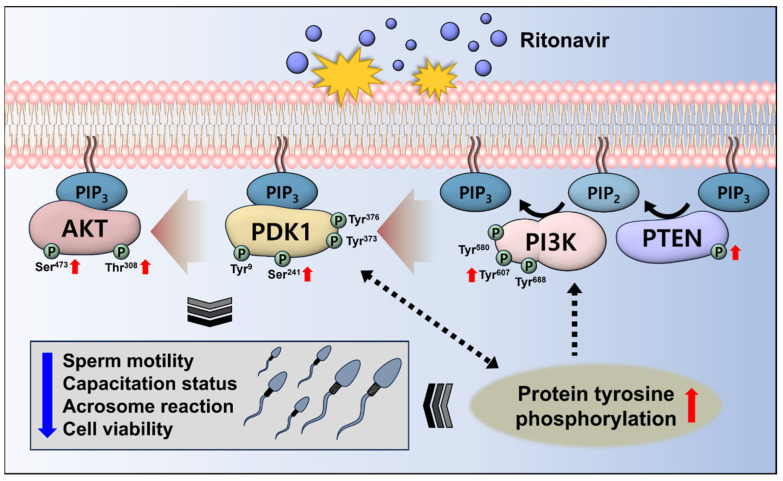
Schematic figure of the PI3K/PDK1/AKT pathway affected by ritonavir (RTV).

**Table 1 toxics-12-00073-t001:** Impacts of ritonavir (RTV) on boar sperm motility and motion kinematics.

Concentration of RTV (µM)					
	Control	0.1	1	10	100
MOT (%)	77.87 ± 1.60 ^A^	78.48 ± 2.22 ^A^	76.73 ± 2.13 ^A^	71.86 ± 2.61 ^A^	45.63 ± 2.97 ^B^
RPD (%)	65.47 ± 2.36 ^A^	65.19 ± 2.92 ^A^	62.78 ± 2.69 ^A^	56.60 ± 3.77 ^A^	32.20 ± 2.38 ^B^
MED (%)	8.19 ± 0.67	8.83 ± 0.70	8.37 ± 0.71	9.48 ± 0.73	8.62 ± 0.57
SLW (%)	3.90 ± 0.58	4.08 ± 0.53	5.25 ± 0.80	5.66 ± 0.95	4.47 ± 0.47
PRG (%)	73.66 ± 2.07 ^A^	74.02 ± 2.64 ^A^	71.15 ± 2.49 ^A^	66.08 ± 3.24 ^A^	40.82 ± 2.91 ^B^
VCL (μm/s)	84.87 ± 3.66 ^A^	87.52 ± 5.18 ^A^	80.91 ± 3.90 ^A^	71.45 ± 4.78 ^A^	38.49 ± 3.23 ^B^
VSL (μm/s)	45.95 ± 2.09 ^A^	44.60 ± 2.01 ^A^	42.00 ± 2.14 ^A^	39.63 ± 2.25 ^A^	20.87 ± 1.31 ^B^
VAP (μm/s)	60.58 ± 2.81 ^A^	60.14 ± 3.29 ^A^	55.57 ± 3.00 ^A^	50.68 ± 3.52 ^A^	26.76 ± 2.09 ^B^
LIN (%)	42.96 ± 1.67 ^A^	40.73 ± 1.35 ^A^	40.10 ± 1.73 ^A^	39.74 ± 1.35 ^A^	24.96 ± 1.30 ^B^
STR (%)	76.22 ± 2.18	74.54 ± 1.69	76.00 ± 1.85	78.79 ± 1.59	78.72 ± 1.80
BCF (Hz)	6.59 ± 0.26 ^A^	6.54 ± 0.30 ^A^	6.48 ± 0.22 ^A^	5.83 ± 0.33 ^A^	2.78 ± 0.18 ^B^
MAD (degree)	64.57 ± 2.91 ^A^	63.42 ± 2.37 ^A^	59.78 ± 2.50 ^A^	57.31 ± 2.77 ^A^	31.49 ± 1.69 ^B^
WOB (%)	71.49 ± 1.43	69.12 ± 1.30	68.66 ± 1.22	70.87 ± 0.62	69.84 ± 1.10
DNC (μm^2^/s)	261.99 ± 24.25 ^A^	291.66 ± 37.26 ^A^	248.29 ± 24.44 ^A^	190.24 ± 26.57 ^A^	62.92 ± 11.47 ^B^
DNM (μm)	7.12 ± 0.51 ^A,B^	7.96 ± 0.58 ^A^	7.50 ± 0.50 ^A,B^	6.29 ± 0.33 ^A,B^	5.78 ± 0.40 ^B^
ALH (μm)	2.98 ± 0.17 ^A^	3.17 ± 0.22 ^A^	2.94 ± 0.16 ^A^	2.49 ± 0.19 ^A^	1.43 ± 0.15 ^B^

Sperm motility and motion kinematics are represented as the mean ± SEM; n = 7. MOT, total sperm motility (%); RPD, rapid sperm motility (%); MED, medium sperm motility (%); SLW, slow sperm motility (%); PRG, progressive sperm motility (%); VCL, curvilinear velocity (μm/s); VSL, straight-line velocity (μm/s); VAP, average path velocity (μm/s); LIN, linearity (%); STR, straightness (%); BCF, beat cross frequency (Hz); MAD, mean angular displacement (degree); WOB, wobble (%); DNC, dance (μm^2^/s); DNM, dance mean (μm); ALH, mean amplitude of lateral head displacement (μm). Different superscript characters (^A and B^) in the same row indicate significant differences (*p* < 0.05), as determined by one-way ANOVA.

## Data Availability

Data are contained within the article and Appendix A.

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
