# Peer review of "Ritonavir Has Reproductive Toxicity Depending on Disrupting PI3K/PDK1/AKT Signaling Pathway"

_toxics, 2024, doi:10.3390/toxics12010073_

Round 1
Reviewer 1 Report
Comments and Suggestions for Authors
The study presented by Ju et al., adds knowledge to mechanisms and effects of the antiviral RTV on sperm cell function. To this reviewer, the study can be published in the present form
Author Response
Reviewer #1: The study presented by Ju et al., adds knowledge to mechanisms and effects of the antiviral RTV on sperm cell function. To this reviewer, the study can be published in the present form.
Response: We would like to express our sincere appreciation for the reviewer’s efforts to review our manuscript.
Reviewer 2 Report
Comments and Suggestions for Authors
The study delving into the reproductive toxicity of Ritonavir (RTV) on the PI3K/PDK1/AKT pathway in spermatozoa makes a noteworthy and valuable contribution to existing literature. Investigating the impact of RTV, a widely used antiviral medication with recognized anti-cancer effects, on sperm functions, the research offers a unique perspective on the potential reproductive implications associated with RTV use.
Despite the substantial contributions, the clarity of the writing needs further refinement. Certain sections require more precise articulation and organization to enhance overall readability. This is essential to ensure that the significance of the study is effectively communicated to a broader audience. A more refined writing style will facilitate a clearer understanding of the research findings and their potential implications. With further modification in the writing, this study has the potential to become an even more impactful and accessible contribution to the field, fostering advancements in our comprehension of RTV's effects on male reproductive health.
Abstract:
· In the sentence "Sperm functions and expression levels of tyrosine-phosphorylated proteins and PI3K/PDK1/AKT pathway-related proteins were evaluated," consider specifying which sperm functions were assessed for clarity.
· The sentence "Additionally, RTV significantly increased levels of phospho-tyrosine proteins and PI3K/PDK1/AKT pathway-related proteins except for AKT and PI3K" might benefit from clarification. If AKT and PI3K levels did not change significantly, it would be helpful to explicitly state this.
· The phrase "Therefore, we suggest that male reproductive toxicity of RTV needs to be recognized by those using or prescribing it" could be made more specific. For instance, you might specify what kind of recognition or precautions should be taken by users or prescribers.
Introduction:
· Clarify the significance of the study and the motivation for investigating the reproductive toxicity of RTV. Why is it important to understand the impact of RTV on the PI3K/PDK1/AKT pathway in spermatozoa?
· Provide a brief overview of the existing knowledge on the topic, especially regarding the effects of RTV on various cell types and its anti-cancer properties.
· Consider rephrasing the sentence: "Although RTV has been broadly used, it induces cell damage in various types of cells including necrosis, mitochondrial damage, endoplasmic reticulum stress, and apoptosis if prescribed at concentrations between 5 and 160 μM [8,9]." for better clarity. You might want to separate the different types of cell damage for a more organized presentation.
· The phrase "reproductive toxicity of RTV in spermatozoa was observed at a lower concentration than that is toxic to other cell types [10]" is somewhat repetitive. Consider rephrasing for smoother flow.
· The transition between discussing the general effects of RTV on cells and specifically on spermatozoa could be smoother. You might consider a sentence that explicitly introduces the focus on sperm functions.
· In the sentence "The ejaculated sperm go through a process called capacitation in the female genital tract to get full capacity for successful sperm-egg fusion [11,12]," consider using a more precise term than "get full capacity." Perhaps, "attain full competence" or "acquire the ability."
· The sentence "Treatment with RTV induces physiological suppression by altering the level of AKT phosphorylation [10]" could benefit from a brief explanation of what is meant by "physiological suppression" for better understanding.
· The phrase "Five semen samples were mixed to avoid individual factors for each replicate" might benefit from clarification. Consider briefly explaining why mixing samples helps avoid individual factors and why this specific number (five) was chosen.
· In the sperm washing procedure, provide details about the rationale or purpose of using a Percoll gradient and washing with BM. Clarify if the washing steps are intended to remove contaminants or unwanted substances.
· Clarify the specific concentrations of RTV used in the capacitation step. You mention that various concentrations were used but don't specify them in this section.
Materials and Methods:
· Specify the rationale for choosing the concentrations of RTV used in the study. Why were concentrations ranging from 0.1 to 100 μM selected?
· Clarify the duration of incubation for capacitation after RTV treatment. This information is essential for understanding the experimental timeline.
· Mention the sample size for each experimental group and the number of replicates conducted. Including this information enhances the transparency and reproducibility of the study.
Results:
· “This section may be divided by subheadings. It should provide a concise and precise description of the experimental results, their interpretation, as well as the experimental conclusions that can be drawn.” Why this has been written in the results section, rather than you follow it for modifying your results section? For better readability, you might consider organizing the presentation of results in a more structured way. Grouping similar findings together or using subheadings for different aspects of the results could enhance clarity.
· In the sentence "AKT and phospho-AKT (Thr308) was detected at approximately 60 kDa and phospho-AKT (Ser473) was detected at approximately 60 and 58 kDa (Figure 3E)," consider using "were" instead of "was" for consistency since you are referring to multiple entities.
· Some sentences could be rephrased for conciseness without losing clarity. For example, "There was no significant change in the expression level of AKT, but a significant increase in the expression level of phospho-AKT (Thr308 and Ser473) was recorded at the highest concentration of RTV (100 μM)" could be simplified for smoother reading.
Discussion:
· In some sentences, consider being more precise in your statements. For instance, in the sentence "RTV, which has been used to control HIV and is now also used as a component of COVID-19 treatment, acts as a booster by inhibiting the metabolism of another reagent [23]," you may want to specify the reagent being referred to for clarity.
· Ensure consistent use of terminology. In one part of the discussion, you mention "abnormal tyrosine phosphorylation," and later you refer to "abnormally increasing tyrosine phosphorylation." Clarify if these terms are used interchangeably or if there is a nuanced difference.
· Ensure a smooth transition between different sections of the discussion. For example, when discussing the altered expression levels of PI3K/PDK1/AKT pathway-related proteins, you could explicitly connect this to the observed effects on sperm functions.
· Be cautious about implying causation based solely on observed correlations. While the study demonstrates associations between RTV treatment, altered signaling proteins, and sperm function, it's crucial to acknowledge that correlation does not imply causation. Further mechanistic studies would be needed to establish causal relationships.
· In the last paragraph, where you refer to Figure 4, it would be beneficial to explicitly state what is depicted in Figure 4 to guide the reader's understanding.
Conclusion:
· It would be helpful to explicitly state the novelty or contribution of your study. For example, you mentioned that it is the first study to investigate the effect of RTV on the PI3K/PDK1/AKT pathway in spermatozoa. Elaborate on why this investigation is significant or how it fills a gap in the existing literature.
· Offer a brief outlook on potential future research directions or applications stemming from the findings. For instance, how might the insights gained from this study contribute to the development of targeted therapies or diagnostics for male infertility?

Author Response
Reviewer #2: The study delving into the reproductive toxicity of Ritonavir (RTV) on the PI3K/PDK1/AKT pathway in spermatozoa makes a noteworthy and valuable contribution to existing literature. Investigating the impact of RTV, a widely used antiviral medication with recognized anti-cancer effects, on sperm functions, the research offers a unique perspective on the potential reproductive implications associated with RTV use.
Despite the substantial contributions, the clarity of the writing needs further refinement. Certain sections require more precise articulation and organization to enhance overall readability. This is essential to ensure that the significance of the study is effectively communicated to a broader audience. A more refined writing style will facilitate a clearer understanding of the research findings and their potential implications. With further modification in the writing, this study has the potential to become an even more impactful and accessible contribution to the field, fostering advancements in our comprehension of RTV's effects on male reproductive health.
Response: We would like to express our sincere appreciation for the reviewer’s efforts to review our manuscript. We read the reviewer’s comments and revised them cautiously.
Abstract:
- In the sentence "Sperm functions and expression levels of tyrosine-phosphorylated proteins and PI3K/PDK1/AKT pathway-related proteins were evaluated," consider specifying which sperm functions were assessed for clarity.
Response: Thank you for your comment. We assessed the sperm functions including sperm motility, motion kinematics, capacitation status, and cell viability. We specified these detailed items in the manuscript (Please see lines 22-24).
- The sentence "Additionally, RTV significantly increased levels of phospho-tyrosine proteins and PI3K/PDK1/AKT pathway-related proteins except for AKT and PI3K" might benefit from clarification. If AKT and PI3K levels did not change significantly, it would be helpful to explicitly state this.
Response: Thank you for your comment. We have stated the alteration of expression levels of AKT and PI3K (Please see line 27-28).
- The phrase "Therefore, we suggest that male reproductive toxicity of RTV needs to be recognized by those using or prescribing it" could be made more specific. For instance, you might specify what kind of recognition or precautions should be taken by users or prescribers.
Response: Thank you for your comment. We have modified the sentence according to the comments of the reviewer (Please see lines 30-31).
Introduction:
- Clarify the significance of the study and the motivation for investigating the reproductive toxicity of RTV. Why is it important to understand the impact of RTV on the PI3K/PDK1/AKT pathway in spermatozoa?
Response: Thank you for your careful review and comments. RTV is known to have cytotoxicity by affecting the PI3K/PDK1/AKT signaling pathway, especially suppressing the phosphorylation of AKT in several cells [Srirangam A et al., Clin Cancer Res, 2006; Batchu RB., Pharmaceuticals, 2014]. Interestingly, however, AKT phosphorylation was significantly increased, and the sperm function was suppressed at the same time by RTV in spermatozoa in our previous study. Nevertheless, since it is still necessary to identify how RTV affects AKT phosphorylation, the present study can be utilized as fundamental research of the molecular mechanism of how RTV affects the PI3K/PDK1/AKT signaling pathway in spermatozoa. We modified the manuscript according to the reviewer’s comments (Please see lines 64-65).
- Provide a brief overview of the existing knowledge on the topic, especially regarding the effects of RTV on various cell types and its anti-cancer properties.
Response: Thank you for your comment. Following the suggestion of the reviewer, we added an overview of the effect of RTV. We added specific explanations about the effects of RTV on normal cells and anti-cancer effects on each cancer cell (Please see lines 39-45).
- Consider rephrasing the sentence: "Although RTV has been broadly used, it induces cell damage in various types of cells including necrosis, mitochondrial damage, endoplasmic reticulum stress, and apoptosis if prescribed at concentrations between 5 and 160 μM [8,9]." for better clarity. You might want to separate the different types of cell damage for a more organized presentation.
Response: Thank you for your comment. Following the suggestion of the reviewer, the effects of ritonavir on normal cells were explained separately according to the type of cell (Please see lines 42-47).
- The phrase "reproductive toxicity of RTV in spermatozoa was observed at a lower concentration than that is toxic to other cell types [10]" is somewhat repetitive. Consider rephrasing for smoother flow.
- The transition between discussing the general effects of RTV on cells and specifically on spermatozoa could be smoother. You might consider a sentence that explicitly introduces the focus on sperm functions.
Response: Thank you for your comment. The repetitive expressions have been modified for smoother expressions, focusing on sperm functions (Please see lines 45-47).
- In the sentence "The ejaculated sperm go through a process called capacitation in the female genital tract to get full capacity for successful sperm-egg fusion [11,12]," consider using a more precise term than "get full capacity." Perhaps, "attain full competence" or "acquire the ability."
Response: Thank you for your comment. The wording has been changed according to the reviewer's comment (Please see line 49).
- The sentence "Treatment with RTV induces physiological suppression by altering the level of AKT phosphorylation [10]" could benefit from a brief explanation of what is meant by "physiological suppression" for better understanding.
Response: Thank you for your comment. The ambiguous expression “physiological suppression” has been deleted, and the sentence was modified to be concise (Please see lines 60-61). The detailed things that occurred due to the phosphorylation of AKT by RTV in sperm were specifically specified in lines 45-47.
- The phrase "Five semen samples were mixed to avoid individual factors for each replicate" might benefit from clarification. Consider briefly explaining why mixing samples helps avoid individual factors and why this specific number (five) was chosen.
Response: Thank you for your comment. Pooling semen can reduce or even eliminate inherent differences in fertility found between sperm and between boars [ALTHOUSE, GARY C., Current Therapy in Large Animal Theriogenology || Artificial Insemination in Swine: Boar Stud Management, 2007]. Therefore, we mixed five semen samples following the previous studies [Lee et al., Reprod Toxicol, 2022]. The draft manuscript has been modified, and the reference has been added (Please see lines 81-83).
- In the sperm washing procedure, provide details about the rationale or purpose of using a Percoll gradient and washing with BM. Clarify if the washing steps are intended to remove contaminants or unwanted substances.
Response: Thank you for your comment. We completely agree with the reviewer's suggestions. We used the Percoll gradient to remove seminal plasma and to obtain motile spermatozoa. After that, the semen sample was washed again to remove the remaining Percoll. This content has been added to the draft manuscript (Please see lines 85-87).
- Clarify the specific concentrations of RTV used in the capacitation step. You mention that various concentrations were used but don't specify them in this section.
Response: Thank you for your comment. The final concentrations of RTV have been added at line 88 in the draft manuscript.
Materials and Methods:
- Specify the rationale for choosing the concentrations of RTV used in the study. Why were concentrations ranging from 0.1 to 100 μM selected?
Response: Thanks for your comments. As references that are added in the draft manuscript, we referred to studies that evaluated the effect of ritonavir treatment on various other cells. In previous studies, RTV was treated at 0 to30 μM to Wistar rats astrocytes in the study about the effects of RTV on viability and glutathione metabolism in brain cells [C. Arend et al., Neurochem Res, 2013]. In addition, 0 to 50 μM, 0 to 100 μM, or 0 to 300 μM of ritonavir were treated in human monocytes and neurons in the study about the effect of antiretroviral therapy on amyloid-β peptide clearance [Lan X et al., J Neuroimmune Pharmacol, 2012]. Moreover, 0 to 100 μM RTV was treated in human ovarian cancer cells in the study about RTV’s anti-cancer effects [S. Kumar et al., Mol Cancer, 2009]. Therefore, we set 0 to 100 μM of ritonavir in our experiments. More references were added to the revised manuscript (Modified in lines 89-90).
- Clarify the duration of incubation for capacitation after RTV treatment. This information is essential for understanding the experimental timeline.
Response: Thank you for your comment. The duration of incubation for inducing capacitation after treating RTV is mentioned in line 88.
- Mention the sample size for each experimental group and the number of replicates conducted. Including this information enhances the transparency and reproducibility of the study.
Response: Thank you for your comment. Each experiment was performed at least 3 times (sperm motility and motion kinematics, 7 times; capacitation status, 8 times; cell viability, 3 times; western blotting, 3 times). The sample size has been mentioned in line 151, and the number of replicates of the experiment is mentioned in each figure legend.
Results:
- “This section may be divided by subheadings. It should provide a concise and precise description of the experimental results, their interpretation, as well as the experimental conclusions that can be drawn.” Why this has been written in the results section, rather than you follow it for modifying your results section? For better readability, you might consider organizing the presentation of results in a more structured way. Grouping similar findings together or using subheadings for different aspects of the results could enhance clarity.
Response: Thank you for your comment. We apologize for confusing the reviewer. The paragraph mentioned by the reviewer was not deleted during the editing process. The paragraph has been deleted. We apologize and thank you once again.
- In the sentence "AKT and phospho-AKT (Thr308) was detected at approximately 60 kDa and phospho-AKT (Ser473) was detected at approximately 60 and 58 kDa (Figure 3E)," consider using "were" instead of "was" for consistency since you are referring to multiple entities.
Response: Thank you for your careful review and comment. The grammatical error has been corrected according to the comment of the reviewer (Please see line 188).
- Some sentences could be rephrased for conciseness without losing clarity. For example, "There was no significant change in the expression level of AKT, but a significant increase in the expression level of phospho-AKT (Thr308 and Ser473) was recorded at the highest concentration of RTV (100 μM)" could be simplified for smoother reading.
Response: Thank you for your comment. The manuscript has been modified according to the reviewer’s comments (Please see lines 190-195).
Discussion:
- In some sentences, consider being more precise in your statements. For instance, in the sentence "RTV, which has been used to control HIV and is now also used as a component of COVID-19 treatment, acts as a booster by inhibiting the metabolism of another reagent [23]," you may want to specify the reagent being referred to for clarity.
Response: Thank you for your comment. The manuscript has been modified according to the reviewer’s comments (Please see lines 226-227).
- Ensure consistent use of terminology. In one part of the discussion, you mention "abnormal tyrosine phosphorylation," and later you refer to "abnormally increasing tyrosine phosphorylation." Clarify if these terms are used interchangeably or if there is a nuanced difference.
Response: Thank you for your careful review and comment. We apologize for confusing the reviewer. There is no nuanced difference between the two expressions that the reviewer pointed out. The terms mentioned by the reviewer have been unified for clarity of the manuscript (Please see lines 231, 289, 308, and 310).
- Ensure a smooth transition between different sections of the discussion. For example, when discussing the altered expression levels of PI3K/PDK1/AKT pathway-related proteins, you could explicitly connect this to the observed effects on sperm functions.
Response: Thank you for your comment. The manuscript has been modified according to the reviewer’s comments (Please see lines 247-248).
- Be cautious about implying causation based solely on observed correlations. While the study demonstrates associations between RTV treatment, altered signaling proteins, and sperm function, it's crucial to acknowledge that correlation does not imply causation. Further mechanistic studies would be needed to establish causal relationships.
Response: Thank you for your careful review and comment. We completely agree with the reviewer. Additional research is planned to establish a causal relationship. A sentence stating that further research is necessary has been added to the manuscript (Please see lines 285-287).
- In the last paragraph, where you refer to Figure 4, it would be beneficial to explicitly state what is depicted in Figure 4 to guide the reader's understanding.
Response: Thank you for your comment. An explanation of Figure 4 has been added to the draft manuscript (Please see lines 306-312).
Conclusion:
- It would be helpful to explicitly state the novelty or contribution of your study. For example, you mentioned that it is the first study to investigate the effect of RTV on the PI3K/PDK1/AKT pathway in spermatozoa. Elaborate on why this investigation is significant or how it fills a gap in the existing literature.
Response: Thank you for your careful review and comment. We completely agree with the reviewer's suggestions. The manuscript has been modified according to the reviewer’s comment (Please see lines 318-320).
- Offer a brief outlook on potential future research directions or applications stemming from the findings. For instance, how might the insights gained from this study contribute to the development of targeted therapies or diagnostics for male infertility?
Response: Thank you for your careful review and comment. After cautious consideration of the reviewer’s comment, mentioned content was deleted because it was judged to be unclear and would not be helpful to readers' understanding.
Reviewer 3 Report
Comments and Suggestions for Authors
As an anti-viral medicine, Ritonavir (RTV) was used to treat COVID-19 disease. It's documented that the pharmacological effects of RTV is mediated by suppressing AKT signaling, with a toxicity to male reproduction system. The abnormal AKT activity might cause disorder of the spermatogenesis, however, its mechanism is still unclear. The authors tackled this question used boar sperm. Overall, the manuscript is well organized and written in fluent English. The following issues need to be checked carefully.
1. Delete the following sentence "This section may be divided by subheadings. It should provide a concise and precise 165 description of the experimental results, their interpretation, as well as the experimental 166 conclusions that can be drawn." in "3. Results".
2. For the capacitation status of the sampel sperm, it's better to provide a few representative images for the four patterns.
Comments on the Quality of English LanguageOverall, the English expression is fine, but careful inspection are still suggested.
Author Response
eviewer #3: As an anti-viral medicine, Ritonavir (RTV) was used to treat COVID-19 disease. It's documented that the pharmacological effects of RTV is mediated by suppressing AKT signaling, with a toxicity to male reproduction system. The abnormal AKT activity might cause disorder of the spermatogenesis, however, its mechanism is still unclear. The authors tackled this question used boar sperm. Overall, the manuscript is well organized and written in fluent English. The following issues need to be checked carefully.
Response: We would like to express our sincere appreciation for the reviewer’s efforts to review our manuscript. We read the reviewer’s comments and revised them cautiously.
- Delete the following sentence "This section may be divided by subheadings. It should provide a concise and precise 165 description of the experimental results, their interpretation, as well as the experimental 166 conclusions that can be drawn." in "3. Results".
Response: Thank you for your comment. We sincerely apologize for mistakes made during the editing process. This sentence has been deleted from the manuscript.
- For the capacitation status of the sampel sperm, it's better to provide a few representative images for the four patterns.
Response: Thank you for your comment. We completely agree with the reviewer. We added the representative images for the four patterns of capacitation status in Supplementary Figure 1.
Reviewer 4 Report
Comments and Suggestions for Authors
The manuscript is devoted to molecular mechanism of toxicity of the protease inhibitor Ritonavir (RTV) suppressing sperm functions by altering protein-kinase B/AKT signaling pathway activity. Based on the results presented by the authors the hypothesis that RTV may suppress sperm functions by induced alterations of the PI3K/PDK1/AKT pathway through abnormally increased tyrosine phosphorylation was confirmed. The significat effect of RTV on capacitation status and cell viability, expecially after treatment with highest concentrations of RTV, changing expression levels of tyrosine-phosphorylated proteins and PI3K/PDK1/AKT pathway related proteins was revealed. Authors indicated that the results presented in the manuscript regarding Ritonavir impact on boar sperm functions that were suppressed appeared similar to their previous experiment with mouse spermatozoa. However, the mechanism of molecular changes in the previuos study remained unclear. Combaining results of the current study with new findings of others research groups authors suggested schematic PI3K/PDK1/AKT signaling pathway indicating intracellular changes that may suppress the sperm functions. To mine opinion their suggestion that in case of administration of RTV, particular attention should be given to male reproductive toxicity, is well-founded.
Author Response
Reviewer #4: The manuscript is devoted to molecular mechanism of toxicity of the protease inhibitor Ritonavir (RTV) suppressing sperm functions by altering protein-kinase B/AKT signaling pathway activity. Based on the results presented by the authors the hypothesis that RTV may suppress sperm functions by induced alterations of the PI3K/PDK1/AKT pathway through abnormally increased tyrosine phosphorylation was confirmed. The significat effect of RTV on capacitation status and cell viability, expecially after treatment with highest concentrations of RTV, changing expression levels of tyrosine-phosphorylated proteins and PI3K/PDK1/AKT pathway related proteins was revealed. Authors indicated that the results presented in the manuscript regarding Ritonavir impact on boar sperm functions that were suppressed appeared similar to their previous experiment with mouse spermatozoa. However, the mechanism of molecular changes in the previuos study remained unclear. Combaining results of the current study with new findings of others research groups authors suggested schematic PI3K/PDK1/AKT signaling pathway indicating intracellular changes that may suppress the sperm functions. To mine opinion their suggestion that in case of administration of RTV, particular attention should be given to male reproductive toxicity, is well-founded.
Response: We would like to express our sincere appreciation for the reviewer’s efforts to review our manuscript.
Round 2
Reviewer 2 Report
Comments and Suggestions for Authors
The reviewer has carefully responded and modified all the raised questions.
Reviewer 3 Report
Comments and Suggestions for Authors
The authors replied my concern point by point and the quality of the manuscript has been greatly improved. I have no more questions.